# Organised Versus Opportunistic Cervical Cancer Screening in Urban and Rural Regions of Lithuania

**DOI:** 10.3390/medicina55090570

**Published:** 2019-09-06

**Authors:** Justina Paulauskiene, Rugile Ivanauskiene, Erika Skrodeniene, Janina Petkeviciene

**Affiliations:** 1Faculty of Public Health, Medical Academy, Lithuanian University of Health Sciences, LT 47181 Kaunas, Lithuania (R.I.); janina.petkeviciene@lsmuni.lt (J.P.); 2Department of Laboratory Medicine, Medical Academy, Lithuanian University of Health Sciences, LT 44307 Kaunas, Lithuania

**Keywords:** cervical cancer, screening, place of residence, non-attendance, barriers, Lithuania

## Abstract

*Background and Objectives*: In 2004, Lithuania started the Nationwide Cervical Cancer Screening Programme. However, screening is more opportunistic than population-wide and the programme’s coverage is insufficient. The aim of this study was to assess the effect of systematic personal invitation on coverage of cervical cancer (CC) screening in urban and rural regions of Lithuania. *Materials and Methods*: The study was conducted in an urban primary healthcare centre (PHCC) and in a rural PHCC, where prevailing CC screening practice was highly opportunistic. Over the first year, all women aged 25–60 who had not received a Pap smear test within the last three years in urban (*n* = 1591) and rural (*n* = 1843) PHCCs received a personal invitation letter to participate in the screening. Over the second year, the reminder letter was sent to the non-attendees (*n* = 1042 in urban and *n* = 929 in rural PHCCs). A random sample of women (*n* = 93), who did not attend for screening after two letters, was contacted by phone in order to identify the barriers of non-attendance. *Results*: Before the study, only 9.6% of the target population in urban and 14.7% in rural PHCCs participated in CC screening. After the first invitation letter, the participation in CC screening increased up to 24.6% in urban and 30.8% in rural areas (*p* < 0.001). After the reminder letter, the attendance was 16.4% in urban and 22.2% in rural PHCCs (*p* < 0.001). The most common barriers for the non-attendance were lack of time, long waiting time for family doctor’s appointment, worries that a Pap test might be unpleasant and preventive gynaecological examination outside of the screening program. *Conclusions*: A systematic personal invitation with one reminder letter significantly increased the coverage of CC screening and was more effective in rural regions than in urban regions. The assessed barriers for non-attendance can be used to improve the coverage of screening.

## 1. Introduction

Globally, cervical cancer (CC) is one of the most common cancers in women. According to GLOBOCAN 2018 estimates, this disease ranks as the fourth most frequently diagnosed cancer and the fourth leading cause of cancer death in the female population worldwide [1]. The highest incidences of CC are reported in Africa, South-Eastern Asia and Eastern Europe, including Lithuania. In 2018, the estimated age-standardized rate of CC incidence was 20/100,000 women in Central and Eastern Europe and 8.5/100,000 women in Western Europe [2]. Lithuania, with an estimated CC incidence rate of 23.7/100,000 women, was among the European countries with the highest CC incidences.

Over the last few decades, cervical cancer incidences and mortality rates have declined in many European populations [3]. This decline is associated with the implementation of population-based cytological screening programmes [4,5]. The sufficient coverage of the target population is one of the most important factors influencing the effectiveness of a CC screening programme. Evidence suggests that well-organised population-based screening strategies and personalised invitation approaches show consistent benefits over opportunistic approaches to CC screening [6,7,8]. Opportunistic screening depends on the initiative of an individual and/or a health advisor. In cervical cancer opportunistic screening settings, a Papsmear test can be offered during contact between a woman and a doctor. The whole target population is not systematically invited, and the screening coverage depends on the frequency of visits to a doctor. The organised screening programmes ensure greater coverage and cost-effectiveness compared with opportunistic screening. According to the European guidelines, the CC screening should only be provided in organised settings [4].

Even in countries with organised CC screening, some women do not attend. A variety of attitudinal, emotional and practical barriers for non-attendance have been described [9,10,11]. There are some differences between countries and within countries in reported barriers to CC screening attendance. Identifying such barriers may help to find ways to increase screening uptake.

In Lithuania, a Nationwide Cervical Cancer Screening Programme was launched in 2004, targeting all women aged 25–60 years [12,13]. The programme is financed by the National Health Insurance Fund under the Ministry of Health of Lithuania and offers a conventional Pap smear test within a three-year interval. The primary healthcare centres are responsible for inviting the women and taking a Pap smear test. The national guidance allows each clinic to provide the diversity of invitation methods, including a verbal invitation during the doctor’s visit or by phone, a written postal invitation or an invitation by an SMS text message. A personal invitation letter with screening details to women of the target population is still a rare practice. Thus, CC screening is more opportunistic and does not assure adequate coverage and participation rate. To increase the CC screening coverage in Lithuania, it is necessary to improve the implementation of the screening programme using evidence-supported strategies.

The pilot project was initiated to move from the opportunistic CC screening to a population-based approach using two systematic personal invitations as recommended by the European Guidelines for CC screening [4]. The aim of this study was to assess the effect of systematic personal invitation on coverage of CC screening in urban and rural regions of Lithuania. We also aimed to determine the barriers of non-attendance in the screening programme.

## 2. Materials and Methods

### 2.1. Study Design and Sample

The study was carried out as a pilot project on the improvement of CC screening coverage in Lithuania by applying a systematic invitation model. In 2014, the Department for Coordination of Preventive Programmes at the Hospital of Lithuanian University of Health Sciences Kaunas Klinikos was established. One of the functions of the Department is the implementation of an organized CC screening programme. The IT database, which helped to perform systematic personal invitation, was created by the Department together with the Kaunas University of Technology using the database of the National Health Insurance Fund. All women identified as unscreened for three or more years are invited personally by a letter to visit a PHCC to have a Pap smear test. In a year, if the women have not had a Pap smear test done, a reminder letter is sent. Both invitation letters include a pre-assigned appointment time and place where a Pap smear test is taken. Conventional cervical smears are read in the Department of Pathology of the Hospital Kauno Klinikos. Cytology results are classified according to the 2001 Bethesda system [14].

The study has been conducted in selected urban (Kaunas) PHCC and rural (Prienai) PHCC. The study protocol was approved by the Lithuanian Bioethics Committee (protocol No BE-2-4, approval date 21 June 2017). Before the study, women in urban and rural PHCCs were invited to participate in CC screening programme by the PHCC provider (a family doctor or a midwife) during appointments which were scheduled for other health issues.

The study was launched on 3 November 2014 in the urban PHCC and on 24 September 2015 in the rural PHCC. Altogether, 3434 women aged 25–60 years, who did not have a Pap smear test within the last three years and were registered at the selected PHCCs, were included in the study. During the first year, all women in both the urban PHCC (*n* = 1591) and rural PHCC (*n* = 1843) received a personal invitation letter by post inviting them to participate in CC screening. After 12 months, non-attendees (1042 in urban PHCC and 929 in rural PHCC) received a reminder letter by post (a second invitation). Personal invitation letter with the pre-fixed appointment time and an information leaflet on CC benefits and harms were prepared and tested by the Department for Coordination of Preventive Programmes according to European Guidelines for Quality Assurance in Cervical Screening [4].

A random sample of women, who did not attend screening after two letters (55 women in the urban PHCC and 38 women in the rural PHCC), were interviewed by phone to identify the barriers for non-attendance for CC screening programme. The questionnaire included 16 ready-made statements about practical, attitudinal and emotional barriers of non-attendance for screening. The women indicated all those which they agreed with. Multiple answers were allowed.

### 2.2. Statistical Analysis

The statistical analysis was performed using IBM SPSS Statistic 20 package. Pearson’s chi-squared test (χ^2^) was used to analyse differences in qualitative variables. A *Z*-test with Bonferroni correction was used for pairwise comparison of proportions. The associations of age and study area with participation rate for CC screening were analysed using multivariate logistic regression analysis. *P*-values of less than 0.05 were considered as statistically significant.

## 3. Results

A year before the beginning of the study, 35.1% (*n* = 574) of women in the urban PHCC and 17.7% (*n* = 383) in the rural PHCC were invited to participate in the CC screening by a family doctor (Figure 1). Only 9.6% (*n* = 169) of women of the target population in urban and 14.7% (*n* = 317) in rural areas attended the screening. After the first invitation letter, the coverage of CC screening increased more in the rural than in the urban PHCC, up to 30.8% and 24.6%, respectively. The lowest participation rate was observed among the women aged 25–34 years, 18.9% in urban and 23.1% rural areas (Table 1). The participation rate after a reminder letter was also significantly higher among rural than the urban female population, 22.2% and 16.9%, respectively. The proportion of women who responded positively to the reminder letter was similar in all age groups.

The multivariate logistic analysis of the associations between study area, age and participation rate in CC screening revealed that odds of participation after the first invitation letter and a reminder letter were by 39% and 44% higher in the rural than the urban PHCC (Table 2). After the first invitation letter, the women of the youngest group (25–34 years) had the lowest odds of attendance for CC screening if compared to all other age groups.

Both invitation letters substantially increased the coverage of CC screening. In the urban PHCC, screening coverage increased from 9.6% after an invitation by a family doctor to 31.8% after the first systematic personal invitation letter and further increased to 41.8% after a reminder letter (Table 3). In the rural PHCC, CC screening coverage increased from 14.7% to 40.9% and 50.5%, respectively.

After the first invitation letter, the proportion of abnormal Pap smear tests was higher in the rural than the urban PHCC at 31.7% and 25.8%, respectively (Table 1). After the reminder letter, the percentage of abnormal cytological tests were similar in both areas at 23.8% and 22.2% respectively. The multivariate logistic regression analysis showed that odds of an abnormal Pap smear test was 39% higher in the rural than in the urban PHCC (Table 2). No association between odds of abnormal Pap smear test and female age was found.

The most common attitudinal and emotional barriers of non-attendance for CC screening in the urban and rural PHCCs were an intention to attend a Pap smear test but facing various obstacles and worries that a Pap smear test might be unpleasant (Table 4). The most common practical barriers for non-attendance in both PHCCs were long waiting times for a family doctor’s appointment and lack of time. A similar proportion of women in urban and rural PHCCs (24.5% and 24.3% respectively) answered that they lack the interest to take a Pap smear test due to regular preventive gynaecological examination and performance of the test outside of screening programme. In a rural area, 24.3% of women noticed that a family doctor did not invite them to participate in the screening programme compared to 11.3% in the urban area. Every fifth woman in rural area answered that the PHCC is too far away from their place of living.

## 4. Discussion

The study was conducted as a pilot project on the improvement of CC screening coverage using a systematic invitation model in urban and rural PHCCs relying on existing opportunistic screening practice in Lithuania. The study showed that a personal invitation letter with one reminder significantly increased the coverage of the CC screening programme in both areas compared with opportunistic screening.

There is enough evidence that an opportunistic screening does not achieve sufficient coverage, which depends on the frequency of visits to a family doctor and the activity of medical personal in providing information about screening. Meanwhile, a well-organized population-based CC screening using personal invitations appears to be an effective method of increasing screening uptake [6,15]. The systematic review published by Ferroni E et al. showed that an invitation letter increased the participation rate up to 50% in CC screening when compared with no intervention [6]. The review of randomised controlled trials of interventions to increase screening uptake also revealed that an invitation letter significantly boosted participation in CC screening (RR = 1.44, 95% CI: 1.24 to 1.52 compared with women who received ordinary healthcare or no invitations) [15]. In our study, participation rates after the first invitation letter reached 24.6% in urban and 30.8% in rural areas, respectively. The earlier study conducted in an urban setting in Lithuania showed a similar attendance rate (27.3%) after a personal invitation letter [16].

A second reminder for non-attendees after the first invitation is crucially important to achieve sufficient screening coverage [17,18]. The systematic review of studies assessing the effectiveness of different invitation methods confirmed that the reminder letter sent to non-attendees after the first invitation significantly improved participation rate [6]. A reminder letter with a scheduled appointment (date, time and place) is likely to be more effective than open reminder letters [19]. In our study, the participation rate after a reminder letter for non-attendees was 16.9% of women in the urban PHCC and 22.2% in the rural PHCC. Similar data were found in the study carried out by Eaker S et al. where attendance after a reminder letter was 15.5% and increased the overall coverage bt 9.2% [17].

In addition to a reminder letter, other invitation methods to improve CC screening uptake have been studied. Broberg G et al. showed that invitation by phone to remind long-term non-attendees significantly increased participation in the screening programme [20]. Eaker S et al. demonstrated that telephone calls after the first letter are also an option to significantly increase female attendance [17]. There is sufficient evidence that offering a self-sampling test for high-risk human papillomavirus types to be done at home is an alternative to reminder letters and an effective method to increase participation in organised CC screening among non-attendees [21]. In Finland, using a self-sampling test as a third intervention after two invitation letters has increased total attendance by 80% [19].

A well-organized population-based CC screening not only enables greater coverage, but also ensures equity and accessibility of each eligible woman of the target population to participate in the screening [4,22,23]. Our data demonstrated higher attendance after the first invitation and reminder letters in rural than urban areas. Overall screening coverage was also higher in the rural PHCC compared to the urban PFCC, 50.5% and 41.8% respectively. These findings are in line with the data from the study carried out in Germany, where the invitation letter was particularly effective among women with lower education migrant women and older women [23].

In our study, the participation in CC screening after the first invitation letter was the lowest in the youngest age group (25–34 years) in both urban and rural areas. After a reminder letter, there was no noticeable difference in attendance between all age groups. A declining rate of participation in CC screening among the women of the youngest age group has been observed in some other countries [24,25]. A decrease in CC screening coverage among young women is of particular concern, as the greatest benefits of screening are achieved when women are screened from a young age. There is some evidence that a reminder letter increases screening coverage among young women [17,25]. Another study showed that a self-sampling test for high-risk human papillomavirus types at home removes some of the barriers and increases attendance among young women [26].

Identifying barriers for CC screening attendance is an important public health issue. The previous Lithuanian study explored the importance of sociodemographic and lifestyle factors for CC screening non-attendance [27]. The study showed that non-attendance was related to a younger age, low education, being single, having rare contacts with a doctor and an unhealthy lifestyle. We analysed practical, attitudinal and emotional barriers. The most common practical barriers for non-attendance were long waiting times for a doctor’s appointment, lack of time and a regular preventive gynaecological examination outside of a screening programme. Inconvenient appointment times were mentioned in the urban area more frequently, while more women in the rural area answered that they had been not invited by a doctor to participate in the screening or the PHCC is too far away from their place of living. The two most common attitudinal and emotional barriers for the non-attendance identified in both areas were the intention to attend a Pap smear test but facing various obstacles and worries that a Pap smear test might be unpleasant.

A range of reasons for non-attendance at CC screening has been analysed by other authors [9,10,11,28]. The comparison of the results is quite problematic because different study methods (study sample, data collection method and questionnaire) were used. Some studies found that practical barriers are more important for screening attendance than emotional barriers [10,11,29]. A lack of time and the difficulty of getting a suitable appointment time were very frequently reported reasons for nonattendance. The non-attendees find it hard to prioritize CC screening before factors related to work, caring for children and family commitments [10,11]. The most effective measure to remove such practical barriers is to offer appointments at different times of the day and week and simplify re-booking. Emotional barriers such as discomfort associated with the gynaecological examination, embarrassment and fear of pain were identified as other important reasons why women did not attend CC screening [9,10,29]. Offering a self-sampling option to non-attendees may help to overcome both practical and emotional barriers to CC screening attendance [29].

Inadequate functioning of a population-based invitation system is one of the weaknesses of the Lithuanian CC screening programme. However, the programme achieved some positive results. The recent study demonstrated the decreasing trends in the CC mortality rate in Lithuania (by 2.3% annually from 2002) [30]. The implementation of organised population-based CC screening is needed for further mortality from CC reduction. Raising public awareness and proper communication strategies would help increase the participation rate. In Lithuania, the National Health Insurance Fund under the Ministry of Health, PHCC and Public Health Bureaus provide some information about the CC screening on their websites, giving lectures, preparing posters and leaflets and organising mass media campaigns.

This study was the first to assess the effectiveness of a systematic personal invitation with a pre-assigned appointment time and a reminder letter for non-attendees to increase CC screening coverage in Lithuania. Moreover, data was collected from both urban and rural areas, enabling us to compare the effect of systematic personal invitation in both settings. Besides, a telephone interview with non-attendees allowed us to assess practical, attitudinal and emotional barriers for non-attendance at CC screening. The findings of this study could contribute to the development of the population-based CC screening in Lithuania.

Some limitations of the study could also be mentioned. First, the study was carried out in only one urban and one rural PHCC. As a result, selection bias was possible. Second, the information about invitation and participation in screening before the pilot study and data on opportunistic smears might be incomplete. Finally, a low number of a telephone interviews, which was related to difficulties to reach non-attendees, did not allow us to examine possible associations between the barriers and sociodemographic characteristics.

## 5. Conclusions

A systematic personal invitation with one reminder letter significantly increases the coverage of CC screening both in urban and rural regions. Our findings provide the evidence for change from an opportunistic to an organised population-based CC screening programme using personal invitation letters. The assessed barriers for non-attendance should be minimized with evidence-based interventions to improve the coverage of screening.

## Figures and Tables

**Figure 1 medicina-55-00570-f001:**
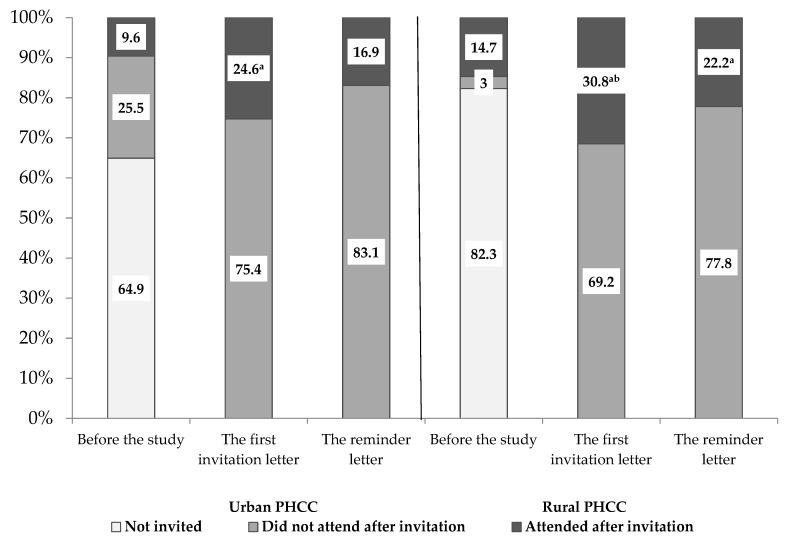
The proportion of invited, non-invited and screened women before the study, after the first personal invitation letter and after reminder letter in urban and rural primary health care centres (PHCC). ^a^
*p* values < 0.001 compared with the coverage of cervical cancer (CC) screening before the study; ^b^
*p* values < 0.001 compared with the coverage of CC screening in Kaunas.

**Table 1 medicina-55-00570-t001:** The participation of women in CC screening by age and the Pap smear test results after the first invitation and reminder letters in urban and rural PHCCs.

	Urban PHCC				Rural PHCC			
	The First Invitation Letter		The Reminder Letter		The First Invitation Letter		The Reminder Letter	
	Invited	Attended	Invited	Attended	Invited	Attended	Invited	Attended
	N	N (%)	N	N (%)	N	N (%)	N	N (%)
**Age (years)**								
25–34	350	66 (18.9) *	245	48 (19.6)	428	99 (23.1) *	221	47 (21.3)
35–44	438	124 (28.3)	281	47 (16.7)	401	143 (35.7)	194	54 (27.8)
45–54	546	129 (23.6)	382	61 (16.0)	619	202 (32.6)	328	72 (22.0)
55–60	257	72 (28.0)	134	20 (14.9)	395	123 (31.1)	186	33 (17.7)
Total	1591	391 (24.6)	1042	176 (16.9)	1843	567 (30.8) **	929	206 (22.2)
**Pap smear test results**								
Unknown result ^a^		20 (5.1)		4 (2.2)		34 (6.0)		5 (2.4)
Normal		270 (69.1)		133 (75.6)		353 (62.3)		152 (73.8)
Abnormal ^b^		101 (25.8)		39 (22.2)		180 (31.7) ***		49 (23.8)

^a^ The Pap smear has been taken but the cytologic test results are unknown; ^b^ Epithelial cell abnormalities: atypical squamous cells (ASC), low grade squamous intraepithelial lesion (LSIL), high grade squamous intraepithelial lesion (HSIL), atypical glandular cell (AGS). * *p* < 0.05 compared with women aged 35–44 and 55–60 in the urban PHCC and women aged 35–44 and 45–54 in the rural PHCC; ** *p* < 0.001 compared with urban women; *** *p* < 0.05 compared with the percentage of abnormal Pap smear tests in the urban PHCC and after a reminder letter in the rural PHCC.

**Table 2 medicina-55-00570-t002:** The odds ratios of participation in CC screening and abnormal Pap smear test results after two systematic invitation letters by study area and age (multivariate logistic regression analysis).

Variable	Participation Rate						Abnormal Pap Smear Tests					
	The First Invitation Letter			The Reminder Letter			The First Invitation Letter			The Reminder Letter		
	OR	95% CI	P	OR	95% CI	P	OR	95% CI	P	OR	95% CI	P
**Study area**												
Urban	1			1			1			1		
Rural	1.39	1.19–1.61	<0.001	1.44	1.15–1.80	0.002	1.39	1.04–1.85	0.027	1.10	0.68–1.78	0.710
**Age (years)**												
25–34	1			1			1			1		
35–44	1.78	1.42–2.23	<0.001	1.08	0.79–1.48	0.630	1.40	0.92–2.13	0.117	1.59	0.80–3.16	0.190
45–54	1.49	1.20–1.84	<0.001	0.90	0.67–1.21	0.501	0.91	0.60–1.37	0.644	1.63	0.85–3.13	0.141
55–60	1.56	1.23–1.99	<0.001	0.75	1.15–1.80	0.122	0.77	0.48–1.23	0.266	1.06	0.44–2.51	0.904

**Table 3 medicina-55-00570-t003:** The effect of systematic personal invitation letters on coverage of cervical cancer screening in urban and rural PHCCs.

	Coverage of Cervical Cancer Screening		
	N	% (95% CI)	Increase %
**Urban PHCC (*N* = 1760)**			
Before the study (invitation by a family doctor)	169	9.6 (8.2–11.0)	-
1st invitation letter	560	31.8 (29.6–34.0)	+231.3
2nd invitation letter	736	41.8 (39.5–44.1)	+31.4
Total increase			+335.4
**Rural PHCC (*N* = 2160)**			
Before the study (invitation by a family doctor)	317	14.7 (13.2–16.2)	-
1st invitation letter	884	40.9 (39.9–42.0)	+178.2
2nd invitation letter	1090	50.5 (49.4–51.5)	+23.5
Total increase			+243.5

**Table 4 medicina-55-00570-t004:** Barriers for nonattendance in a cervical cancer screening programme in urban and rural PHCCs: the phone interview.

	Urban PHCC (*N* = 55)	Rural PHCC (*N* = 38)
**Attitudinal and emotional barriers**	**%**	**%**
Intends to attend for a Pap smear test but faces various obstacles	52.7	36.1
Worries that a Pap smear test might be unpleasant	34.6	36.8
Believes that she is not at risk of cervical cancer	21.2	25.0
Is afraid to be diagnosed with cervical cancer	20.8	27.8
Negative experience during a Pap test in the past	20.0	8.3
Feels healthy and sees no need for a Pap test	15.4	22.2
Sexually inactive for a long time and sees no need to attend	15.1	16.2
Doesn’t trust the efficiency of a Pap test	9.3	13.5
**Practical barriers**	**%**	**%**
The long waiting-time for doctor’s appointment	49.1	34.3
Lack of time due to long working hours or family duties	41.5	29.7
Has a regular gynaecological examination	24.5	24.3
Inconvenient appointment time	23.6	10.8
A family doctor doesn’t invite to participate in the screening	11.3	24.3
Has never heard of a Pap test	7.5	16.2
Has never been invited to have a Pap test	7.5	22.2
The clinic is too far away from women’s living place	1.9	22.2

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
