# Peer review of "Organised Versus Opportunistic Cervical Cancer Screening in Urban and Rural Regions of Lithuania"

_medicina, 2019, doi:10.3390/medicina55090570_

Round 1
Reviewer 1 Report
This is a very-well written manuscript that has been a
pleasure to review. It's findings are clear-cut and in line
with parallel reports in the literature.
I have no hesitation in recommending this work for
publication.
I congratulate these investigators on the work presented.
Author Response
Point 1: This is a very-well written manuscript that has been a pleasure to review. It's findings are clear-cut and in line with parallel reports in the literature. I have no hesitation in recommending this work for publication. I congratulate these investigators on the work presented.
Response 1: Thank you for favourable and encouraging evaluation of our manuscript.
Reviewer 2 Report
Thank you for the opportunity to review this manuscript. This paper reports on a study that aimed to evaluate the effect of a systematic personal invitation to screen for cervical cancer on people in urban and rural regions of Lithuania.
This is a really interesting study that will add to the body of cancer prevention/screening science. Overall, the paper is well written and clear. I do not have any questions about the scientific soundness. However, I think the paper would benefit from a few edits.
1) On page 1, the last sentence in the first paragraph under the introduction, the incidence rate of "23,7/10000" should probably read 23.7/10000.
2) The authors introduce the term "opportunistic" on page 2 in the introduction. I recommend describing what opportunistic screening means and how it is different from the approach used in the study. Readers who are not familiar with population/public health jargon would likely not know what that means.
3) The authors need to provide more information about the interview data. On page 3, the authors describe interviewing the women who did not attend cervical cancer screening. I suggest adding details about the collection of the data (e.g., were the subjects asked open-ended questions or closed-ended questions from a questionnaire?). Also, how was this data analyzed?
4) The discussion section of the paper is excellent. Do the authors have any data on the cost of this intervention? I think an additional discussion piece about the cost of this would attract readers who are looking at strategies to increase screening rates in their own programs.
5) Finally, although the English is good, there are a number of grammatical errors throughout the paper. However, the grammatical errors did not detract from the clarity.
Author Response
Thank you for your helpful comments for our manuscript. We hope that we have successfully addressed all of the concerns raised, and we believe that the manuscript has been substantially improved. Our detailed responses to the comments and the description of the changes we have made to the manuscript are provided below.
Point 1: On page 1, the last sentence in the first paragraph under the introduction, the incidence rate of "23,7/10000" should probably read 23.7/10000.
Response 1: We would like to thank the reviewer for noticing an error. It was corrected.
Point 2: The authors introduce the term "opportunistic" on page 2 in the introduction. I recommend describing what opportunistic screening means and how it is different from the approach used in the study. Readers who are not familiar with population/public health jargon would likely not know what that means.
Response 2: Although we believe that the term “opportunistic screening” is widely known and accepted in public health and medical community, for clearer and more precise paper we included the explanation of the term in the Introduction: “Opportunistic screening depends on the initiative of an individual and/or a health advisor. In cervical cancer opportunistic screening settings, Pub smear test can be offered during contact between a woman and a doctor. The whole target population is not systematically invited and the screening coverage depends on the frequency of visits to a doctor.”
Point 3: The authors need to provide more information about the interview data. On page 3, the authors describe interviewing the women who did not attend cervical cancer screening. I suggest adding details about the collection of the data (e.g., were the subjects asked open-ended questions or closed-ended questions from a questionnaire?). Also, how was this data analyzed?
Response 3: Following the reviewer’s comment, we added to the Methods section additional information about the telephone interview: “The questionnaire included 16 ready-made statements about practical, attitudinal and emotional barriers of non-attendance for screening. The women indicated all those which they agreed with. Multiple answers were allowed.”
Point 4: The discussion section of the paper is excellent. Do the authors have any data on the cost of this intervention? I think an additional discussion piece about the cost of this would attract readers who are looking at strategies to increase screening rates in their programs.
Response 4: This is a very good point. A cost-effectiveness analysis of systematic invitation approach versus opportunistic screening approach in a cervical cancer screening programme is in the process of analysis and preparation of other research publication. To our knowledge, it will be the first attempt to evaluate the cost-effectiveness of different cervical cancer screening approaches in Lithuania.
Point 5: Finally, although English is good, there are a number of grammatical errors throughout the paper. However, the grammatical errors did not detract from the clarity.
Response 5: Thank you for the comment. We do acknowledge our imperfections in the English language, although we put a lot of effort into this question. We asked for some help from a native English speaking colleague who edited the manuscript.
Reviewer 3 Report
This is a good report of the effect of one component of organised screening programs for cervical cancer, the effect of personal invitations and reminders.
In spite of the invitations, the majority of participants failed to attend for screening, with various reasons cited for this. However, it seems likely that many of those who refused had insufficient knowledge of the importance of screening. One important component of organised screening is public, and professional education. It would be helpful if the authors provided information on the extent such programs exist in Lithuania.
Author Response
Thank you for your helpful comment for our manuscript. Our response to the comment and the description of the changes we have made to the manuscript are provided below.
Point 1: In spite of the invitations, the majority of participants failed to attend for screening, with various reasons cited for this. However, it seems that many of those who refused had insufficient knowledge of the importance of screening. One important component of organised screening is public, and professional education. It would be helpful if the authors provided information on the extent such programs exist in Lithuania.
Response 1: Following the reviewer’s advice, we added to the Discussion section: “Raising public awareness and proper communication strategy would help increase the participation rate. In Lithuania, the National Health Insurance Fund under the Ministry of Health, PHCC and Public Health Bureaus provide some information about the CC screening on their websites, giving lectures, preparing posters and leaflets, organising mass media campaigns.”
Round 2
Reviewer 3 Report
None